# The Diurnal Susceptibility of Subtropical Clouds to Aerosols

Marcin J. Kurowski<sup>1</sup>, Matthew D. Lebsock<sup>1</sup>, and Kevin M. Smalley<sup>2</sup>

<sup>1</sup>Jet Propulsion Laboratory, California Institute of Technology, Pasadena, California, USA.

Abstract. The diurnal susceptibility of clouds and their radiative properties to aerosols is examined during their Lagrangian transition from subtropical stratocumulus to shallow cumulus regimes. Using large-eddy simulations, we analyze the six-day evolution of an air mass along a 3,800-km observed trajectory from the coast of Peru toward the equator. Pristine and polluted scenarios are simulated with forcing imposed from weather reanalysis. The polluted scenario exhibits stronger diurnal variations in cloud water, cloud fraction, and albedo, with enhanced nighttime entrainment and suppressed precipitation. The overall response of cloud properties and outgoing shortwave radiation to droplet number concentration follows a distinct diurnal pattern: strong positive cloud adjustments dominate at night and in the morning, while weak negative adjustments prevail in the afternoon. This cycle is driven by the competition between precipitation suppression, which enhances cloud water and coverage, and entrainment drying, which depletes them. In polluted conditions, enhanced entrainment leads to a deeper and more decoupled boundary layer that cannot be sustained by surface fluxes in the afternoon, resulting in negative cloud adjustments. The enhanced entrainment rate under polluted conditions is caused by the reduced sedimentation of cloud and precipitation water from the entrainment zone. While the Twomey effect dominates the diurnal average albedo response, the diurnal variation in the competing cloud adjustments lead to a near-neutral net adjustment effect in the afternoon, highlighting the critical role of diurnally varying processes in aerosol-cloud interactions.

. The authors' copyright for this publication is transferred to Jet Propulsion Laboratory, California Institute of Technology.

# 1 Introduction

The interactions between aerosol and clouds represent one of the largest sources of uncertainties in the anthropogenic radiative forcing of Earth's climate (IPCC, 2021, 2022). The radiative effect of the collective set of changes to cloud morphology by aerosol is known as the Effective Radiative Forcing due to Aerosol Cloud Interactions (ERF<sub>ACI</sub>; Boucher et al., 2013), which is composed of a number of different cloud changes (Twomey, 1977; Albrecht, 1989; Boucher and Lohmann, 1995; Lohmann and Feichter, 2005; Solomon et al., 2007; Wall et al., 2022). The first order effect, often referred to as the Twomey effect (Twomey, 1977) posits that an increase in cloud droplet number ( $N_c$ ) for fixed cloud liquid water path (LWP<sub>c</sub>) results in a greater integrated water droplet cross sectional area and thus an increase in cloud optical depth ( $\tau_c$ ) and cloud albedo ( $A_c$ ). The magnitude of the Twomey effect is thought to be relatively well understood (Fan et al., 2016; Bellouin et al., 2020; Quaas

<sup>&</sup>lt;sup>2</sup>Lawrence Livermore National Laboratory, Livermore, California, USA.

et al., 2020). However, second-order indirect effects, or cloud adjustments, result from changes to the cloud liquid water path (LWP<sub>c</sub>) and cloud cover fraction ( $f_c$ ), where the domain mean liquid water path is LWP =  $f_c$ LWP<sub>c</sub>. These cloud adjustments are less well understood. It was first thought that increases in  $N_c$  would inhibit the formation of precipitation and thus increase cloud lifetime (Albrecht, 1989; Pincus and Baker, 1994). More recently, it was suggested that increasing  $N_c$  can reduce LWP<sub>c</sub> through a decreased sedimentation efficiency causing an increase in liquid near the cloud top which enhances the efficiency of the entrainment of dry free-tropospheric air into the cloud layer (Ackerman et al., 2004; Bretherton et al., 2007).

To quantify the various aerosol-cloud interactions, the sensitivity of the reflected shortwave flux  $(F^{\uparrow})$  is often decomposed into three terms (Bellouin et al., 2020) representing changes in cloud optical depth at fixed LWP<sub>c</sub> (denoted  $S_N$ ), additional changes in  $\tau_c$  resulting from changes in LWP<sub>c</sub> at fixed  $N_c$  (denoted  $S_{LWP}$ ) and changes in  $f_c$  at fixed  $\tau_c$  (denoted  $S_f$ ):

35 
$$\frac{dF^{\uparrow}}{d\ln N_c} = \underbrace{\frac{\partial F^{\uparrow}}{\partial \ln N_c}}_{\text{Twomey Effect }(S_N)} + \underbrace{\frac{\partial F^{\uparrow}}{\partial \ln \text{LWP}_c} \cdot \frac{d\ln \text{LWP}_c}{d\ln N_c}}_{\text{LWP adjustment }(S_{\text{LWP}})} + \underbrace{\frac{\partial F^{\uparrow}}{\partial f_c} \cdot \frac{df_c}{d\ln N_c}}_{\text{Fraction adjustment }(S_f)}. \tag{1}$$

There is observational evidence for both increases and decreases in the LWP<sub>c</sub>. For example, Han et al. (2002) use satellite data to show that clouds have positive, negative, and neutral sensitivity to aerosol in roughly equal proportions. It is also clear that the sign of the response is dependent on the cloud state. Lebsock et al. (2008) find that the LWP<sub>c</sub> tends to increase with increased  $N_c$  for precipitating clouds and decrease with increasing  $N_c$  for non-precipitating clouds. Evidence from ship-tracks show both positive and negative sensitivity (Ackerman et al., 2000; Coakley and Walsh, 2002), with the observation that the sign of the response is associated with the mesoscale cellular structure with open-celled regimes tending to have a positive response and closed-cells tending to have a negative response, presumably due to their differential propensity to precipitate (Christensen and Stephens, 2012). A recent review of polluted clouds down-wind of anthropogenic pollution sources finds a weak albeit slightly negative average response of LWP<sub>c</sub> to aerosol perturbations (Toll et al., 2019). To the contrary, Manshausen et al. (2022) recently find a large positive increases in LWP<sub>c</sub> by using ship location data to find a large number of 'invisible' ship tracks, which are not readily identifiable in satellite imagery. Regional variability and observational uncertainties, such as cloud regime differences, further complicate LWP responses (Wood, 2012). We note that cloud adjustments are dependent on the background  $N_c$ , which can explain the presence of both positive and negative adjustments without contradiction. This state dependence in the cloud adjustment is manifest as the 'inverted V' relationship between  $N_c$  and LWP<sub>c</sub> implying postive adjustment at low  $N_c$  and negative adjustment at high  $N_c$  (Gryspeerdt et al., 2022).

Observed positive correlations between aerosol optical depth and  $f_c$  have long been considered suspect due to the tendency to observe enhanced clear sky reflectance in the vicinity of clouds due to three dimensional radiative effects (Várnai and Marshak, 2009). For example, carefully controlling for the distance of an aerosol retrieval to the nearest cloud nearly halves the magnitude of the relationship between  $f_c$  and aerosol optical depth (Christensen et al., 2017). To entirely avoid the influence of artificial correlations, more recent observational studies have used either the observed  $N_c$  or a model derived aerosol field in place of the aerosol optical depth to derive the slope  $df_c/d\ln N_c$ . Although the magnitude is highly uncertain, studies tend to find a positive correlation (Gryspeerdt et al., 2016; Wall et al., 2023).

Most observational satellite studies are based on visible and near infrared imager data with fixed diurnal sampling time therefore there are few hints as to the observed diurnal cycle of the cloud adjustments. A study of a South Atlantic shipping lane shows that Terra MODIS shows a larger positive LWP adjustment than Aqua MODIS, and the Terra/Aqua show positive/negative  $f_c$  adjustments (Diamond et al., 2020). The recent observational study of Smalley et al. (2024) uses a combination of geostationary and microwave imager data to find a strong diurnal cycle in the response of the domain mean LWP to variation in  $N_c$ . Decreases in LWP are observed during the day and neutral or positive responses of LWP during the night time hours. They speculated that this diurnal cycle in LWP sensitivity was driven primarily by the diurnal variation in precipitation sensitivity, however there is no way to confirm or refute the causation with observations. The discovery of this large diurnal cycle in the cloud adjustments presents yet another significant uncertainty in our current knowledge because the ERF<sub>ACI</sub> is weighted by the diurnally varying incoming solar radiation.

Many Large-Eddy Simulation (LES) studies employ idealized scenarios with constant forcings to extract key controls of the cloud system and simplify the interpretation of the results. This approach has been foundational in studies of aerosol indirect effects, where aerosols modify cloud albedo and lifetime through changes in droplet number and precipitation processes (e.g., Moeng et al., 1996; Feingold et al., 1999; Khairoutdinov and Kogan, 2000; Jiang et al., 2002; Stevens et al., 2005; Lu and Seinfeld, 2005; Hoffmann et al., 2020). While this approach has the advantage of simplicity, it neglects two important modes of variability in the subtropical cloudy boundary layer: (1) the large diurnal cycle, and (2) the multi-day transition of stratocumulus to cumulus boundary layers. A handful of studies have touched on these modes of variability in the context of aerosol indirect effects. For example, the study of Sandu et al. (2008) shows that increases in  $N_c$  increase the amplitude of the diurnal cycle of LWPc in simulated stratocumulus. Furthermore, Sandu and Stevens (2011) show that transitions from Stratocumulus to Cumulus are a response to increasing sea surface temperature (SST) through Lagrangian LES in the North East Pacific. However, Yamaguchi et al. (2017) find that aerosol number concentration influences the timing of the transition through its mediation of drizzle. Prabhakaran et al. (2024) perform Lagrangian LESs of stratocumulus clouds transitioning to cumulus, perturbed by intermittent aerosol injections to simulate marine cloud brightening. They find that aerosol perturbations suppress precipitation and enhance cloud reflectivity, with greater sensitivity in pristine conditions due to precipitation-driven transverse circulations, and note diurnal variations in radiative forcing due to the solar cycle. Zhang et al. (2024) use LES with a conditional Monte-Carlo subsampling approach to study non-precipitating marine stratocumulus, finding a diurnal cycle in cloud property sensitivity where aerosol-induced liquid water path adjustments are more negative at night due to enhanced entrainment, but less negative in the afternoon, buffered by shortwave absorption dependent on cloud liquid water path. Erfani et al. (2022) perform Lagrangian LES of a stratocumulus-to-cumulus transition in a subtropical marine environment, demonstrating that aerosol-induced LWP adjustments depend on the cloud regime. In pristine conditions with active precipitation, aerosol perturbations suppress drizzle, leading to larger LWP increases in stratocumulus clouds compared to polluted conditions, where precipitation is already limited.

This study addresses the susceptibility of clouds and their properties to aerosol concentrations along their realistic multi-day Lagrangian transition from the subtropics to the tropics, with a focus on their diurnal variability. Furthermore, we decompose the susceptibility into three main components: the Twomey effect, LWP adjustment, and cloud fraction adjustment, showing

that the Twomey effect is the primary factor controlling this susceptibility, while the other two contribute notably to the diurnal variability. The combined effects result in a significant susceptibility during the morning hours, with a diminishing net effect in the afternoon and evening. Our methodology is outlined in Section 2, the simulation results and their analysis are presented in Section 3, and Section 4 summarizes the study and presents the conclusions.

## 2 Methodology

100

105

# 2.1 Lagrangian Trajectory

The Lagrangian trajectory used in this study was produced using the methodology outlined in Smalley et al. (2022), and was then selected from the ensemble generated in Smalley et al. (2024). The trajectory is propagated forward in time using a 10 minute time step with the 3-hourly 925 hPa winds from the Modern-Era Retrospective analysis for Research and Applications, version 2 (MERRA-2; Gelaro et al., 2017). The selected trajectory west of Peru spans about 3,800 km and extends from the subtropics to the tropics over the Pacific Ocean (Fig. 1). It represents a classical example of the stratocumulus-to-cumulus transition for an air mass propagating over the ocean upon increasing sea surface temperature and reduced large-scale subsidence. It starts at 20°S and 80°W on 2019-10-06 00:00:00 UTC (i.e., around 18:00 local time) and follows the mean planetary boundary layer (PBL) flow during its six-day evolution. Note that the calculated trajectory provides only an approximate reconstruction of the real air mass movement due to both the presence of wind shear and the limited accuracy and resolution of reanalysis data.

Several observed cloud properties are matched to the trajectory where they are available. These include LWP from the fleet of passive microwave imagers (Wentz and Spencer, 1998). Higher frequency LWP observations are taken from the corrected geostationary data of Smalley and Lebsock (2023). Additional geostationary data products derived from the Advanced Baseline Imager (ABI) on GOES-16 include the cloud fraction, cloud top height, cloud optical depth, and cloud effective radius (Walther and Straka, 2019–2021). Finally, the profiles of several MERRA-2 variables are collocated along the trajectory to provide forcing data for the LES. These variables include horizontal wind components, water vapor, potential temperature, and large-scale subsidence, in addition to sea surface temperature.

# 2.2 Large-Eddy Simulations

We use the System for Atmospheric Modeling (Khairoutdinov and Randall, 2003) to simulate the transition. The domain size is  $40.92 \times 40.92 \text{ km}^2$ . The horizontal grid spacing is 40 m, while the vertical grid spacing is 8 m in the PBL, gradually increasing with altitude. The initial and boundary conditions are based on MERRA-2 reanalysis data interpolated to the trajectory points. However, adjustments to the initial atmospheric state were necessary to reproduce the thick stratocumulus layer observed on that day. The original MERRA-2 profiles, due to their coarse vertical resolution and smoothed inversion structure, only support shallow convection when used in LES. To enable stratocumulus formation, the inversion layer thickness was reduced to around

40 m, providing a sharper capping inversion more consistent with stratocumulus-topped boundary layers (Stevens et al., 2005; Berner et al., 2011).

The free-tropospheric temperature and moisture profiles are nudged with 1-hr timescale starting 500-m above the PBL height defined as the top of inversion layer. Because the model cannot directly follow changes in the mesoscale pressure gradient that controls boundary-layer winds, we apply weak nudging of the mean PBL winds with a timescale of 12 hours. Furthermore, to suppress the development of spurious circulations within the domain during longer simulations, we apply weak horizontal homogenization of temperature and water vapor mixing ratio with a 48-hour timescale. Microphysics is parameterized using the scheme of Khairoutdinov and Kogan (2000). Instead of prognosing cloud droplet number concentrations, four different aerosol-related scenarios are prescribed along the trajectory in terms of fixed time-dependent concentrations (Fig. 1 e). All scenarios begin with high coastal droplet number concentrations typical of polluted continental air, gradually decreasing to 25 cm<sup>-3</sup> for pristine air, 50 cm<sup>-3</sup> and 100 cm<sup>-3</sup> for intermediate conditions, and 200 cm<sup>-3</sup> for polluted air. These scenarios represent realistic variability in number concentrations and the associated aerosol-cloud interactions including their impact on cloud microphysics and radiative properties. The 25 cm<sup>-3</sup> asymptotic case best agrees with the satellite observations and should be considered the baseline simulation. The sea surface temperature changes from approximately 290 K to nearly 297 K, with surface fluxes interactively calculated based on local atmospheric conditions near the surface. Interactive short-wave and long-wave radiation effects are also included. A similar Lagrangian perspective and modeling setup was applied in many other studies (e.g., van der Dussen et al., 2013; Sandu and Stevens, 2011; Yamaguchi et al., 2017). Note that while the boundary conditions follow observations, the PBL development is determined by the processes occurring within it.

Finally, we comment that in this case, the impact of changing subcloud atmospheric conditions on surface moisture supply across different scenarios is relatively small, as latent surface heat fluxes increase by only several percent in the polluted scenario compared to the pristine one, with 6-day averages of 137 and 145  $\rm W\,m^{-2}$ , respectively (see supplemental material). All other simulation results, including sensitivity scenarios analyzed further, fall within that envelope determined by the pristine and polluted scenarios.

# 2.3 Diurnal Controls of Indirect Radiative Effect

125

130

To understand the relative diurnal contributions of the Twomey effect and the cloud adjustments to the indirect effect, offline radiative transfer calculations are performed. The reflected shortwave flux is given by

$$F^{\uparrow} = F_o \mu_o A \,, \tag{2}$$

where  $F_o$  is the solar constant,  $\mu_o$  is the cosine of the solar zenith angle, and A is the all-sky albedo. The all-sky albedo is calculated as the sum of a clear and cloud sky components

$$A = (1 - f_c)\alpha_{surf} + f_c A_c , \qquad (3)$$

where  $\alpha_{surf}$  is the ocean surface albedo assumed to be 0.06, and  $A_c$  is the albedo of the cloudy part of the domain. We neglect clear sky absorption of the radiation. Accounting for multiple reflections between a cloud layer with albedo ( $\alpha_{cld}$ ) and the

155 reflecting surface (Stephens, 1984) gives the combined albedo for the cloudy part of the domain as

$$A_c = \alpha_{cld} + \frac{\alpha_{surf}(1 - \alpha_{cld})^2}{1 - \alpha_{surf}\alpha_{cld}} \,. \tag{4}$$

Appendix A describes the offline calculations of  $\alpha_{cld}$ , including a proper accounting of the solar zenith angle, which is a critical factor when addressing the diurnal cycle. Finally, the Cloud Radiative Effect (*CRE*) is calculated as

$$CRE = F_0 \mu_0 f_c (A_c - \alpha_{surf}) . \tag{5}$$

The cloud optical depth is calculated at each time step from the domain mean time-dependent modeled  $LWP_c$  and  $N_c$  assuming an adiabatic cloud vertical structure (Brenguier et al., 2000) following the specific implementation of Hoffmann et al. (2023)

$$\tau_c = 0.2 N_c^{1/3} LW P_c^{5/6}$$
 (6)

The offline radiation calculations are used to decompose the ERF<sub>ACI</sub> into the three indirect sensitivity terms defined in Eq. 1. Knowing that  $F^{\uparrow} = F^{\uparrow}(N_c, LWP_c, f_c)$ , the sensitivity can be estimated using the pristine and polluted simulation results as follows:

$$S_N = \frac{\partial F^{\uparrow}}{\partial \ln N_c} \approx \frac{F^{\uparrow}(N_{c200}, \overline{\text{LWP}}_c, \overline{f_c}) - F^{\uparrow}(N_{c25}, \overline{\text{LWP}}_c, \overline{f_c})}{\ln N_{c200} - \ln N_{c25}},$$
(7)

$$S_{\text{LWP}} = \frac{\partial F^{\uparrow}}{\partial \ln \text{LWP}_c} \cdot \frac{d \ln \text{LWP}_c}{d \ln N_c} \approx \frac{F^{\uparrow}(\overline{N_c}, \text{LWP}_c(N_{c200}), \overline{f_c}) - F^{\uparrow}(\overline{N_c}, \text{LWP}_c(N_{c25}), \overline{f_c})}{\ln N_{c200} - \ln N_{c25}}, \tag{8}$$

$$S_f = \frac{\partial F^{\uparrow}}{\partial \ln N_c} = \frac{\partial F^{\uparrow}}{\partial f_c} \cdot \frac{df_c}{d \ln N_c} \approx \frac{F^{\uparrow}(\overline{N_c}, \overline{\text{LWP}}_c, f_c(N_{c200})) - F^{\uparrow}(\overline{N_c}, \overline{\text{LWP}}_c, f_c(N_{c25}))}{\ln N_{c200} - \ln N_{c25}} \,. \tag{9}$$

Here, the overbar denotes the mean of the polluted and pristine values along the trajectory as a function of time. This means that for each of the three terms, i.e., Eqs. 7-9, we estimate the sensitivity of  $F^{\uparrow}$  in only one direction within the three-dimensional parameter space  $(N_c, \text{LWP}_c, f_c)$ , while holding the other two parameters fixed at their mean values. Note that because these sensitivities are expressed in terms of reflected fluxes rather than albedo, they inherently account for the diurnal variation in incoming solar radiation and therefore drop to zero at night.

#### 175 3 Results

#### 3.1 Evaluation of LES evolution against observations

We begin by evaluating the diurnal evolution of the simulated clouds and the realism of the LES against the observations. Figure 1 provides a summary of the evolution of the clouds for four  $N_c$  scenarios over the six-day simulation. Panel a shows the path

of the trajectory while panel f shows the sea surface temperature along the trajectory. Panel e shows the imposed number concentrations, loosely based on observations from the ABI, which begins at large values of several hundred cm<sup>-3</sup> near the coast and asymptotes to values ranging between 25 and 200 cm<sup>-3</sup> in the tropics. Most of this paper will contrast the pristine  $(25 \text{ cm}^{-3})$  simulation with the polluted  $(200 \text{ cm}^{-3})$  simulation. Note that the pristine scenario best matches the observations and the polluted scenario should be interpreted as a perturbation from the observed state. Panel h shows the expected increases in  $\tau_c$  with increases in  $N_c$ . Panels d (polluted) and i (pristine) highlight two critical features of the simulations. First, the polluted cloud grows significantly deeper than the pristine cloud and that growth occurs in the overnight and early morning hours. Second, the pristine cloud produces substantially more drizzle than the polluted clouds. Each of these observations is consistent with expectations that increasing  $N_c$  should both suppress precipitation and increase the cloud top entrainment efficiency. Next, note that the diurnal evolution of the LWP and  $f_c$  (panels b, g) show general agreement with the observations, while differing in some of the precise details. For example, the LES is not able to produce sufficiently thick and extensive cloud cover over the nighttime hours of days 2-4. We also note that the pristine experiment, which is the most realistic scenario, is well able to simulate the observed cloud top height (CTH), whereas the more polluted experiments show larger growth of the cloud layer (panel c), which is not observed in this case but remains a physically plausible outcome under different conditions.

## 3.2 Cloud Radiative Effects

How does the distinct diurnal variation in cloud properties affect the ERF<sub>ACI</sub>? Figure 2 contrasts the pristine and polluted scenarios to understand the relative influence of cloud adjustments relative to the Twomey effect on the CRE. The largest differences in LWP<sub>c</sub> occur during the overnight and early morning hours due to the suppression of precipitation (panel a). In contrast, during mid-day, the polluted LWP<sub>c</sub> is smaller than the pristine scenario. The  $f_c$  adjustment follows a similar diurnal pattern, with the polluted scenario showing a larger  $f_c$  overnight into the morning, and a smaller  $f_c$  in the afternoon (panel b). Panel c compares the cloud albedo of the pristine and polluted scenarios, including both the Twomey effect and the LWP adjustment. The polluted  $A_c$  is generally larger than the pristine, except for a few hours during midday when the reductions in LWP<sub>c</sub> more than offset the Twomey effect brightening. What ultimately matters for the energy budget of the system is the CRE shown in panel e. Here, a distinct diurnal pattern emerges in the difference between the polluted and pristine scenarios. In the polluted scenario, there is a distinct increase in CRE in the morning, while in the early afternoon, there are modest decreases. Occasionally, a secondary increase in CRE occurs in the evening when the cloud layer is recovering from its afternoon minimum.

It is important to recognize that two factors limit the sensitivity of CRE to  $N_c$  at large solar zenith angle, i.e., near sunrise and sunset. First, the incoming solar flux scales as  $\mu_o$  and second as the cloud albedo approaches unity the Twomey effect tends to zero. As a result, the fairly large cloud adjustment terms in the early morning hours are not very effective at increasing the diurnal average CRE (see Figure 2d/e).

Figure 3 shows a composite diurnal cycle of the ERF<sub>ACI</sub> averaged over the six-day trajectory. Here panel a shows the three individual terms that determine the ERF<sub>ACI</sub>, calculated from Eqs. 7-9. The Twomey effect  $(S_N)$  is always positive with a peak in the late morning. The timing of this peak in  $S_N$  results from a combination of the fact that sensitivity is maximum for

 $A_c = 0.5$  (Platnick and Twomey, 1994) and of the fact that morning hours have larger  $f_c$  than afternoon hours, so that the Twomey effect has less leverage in the afternoon than in the morning. The other two cloud adjustment terms ( $S_{LWP}$ ,  $S_f$ ) have similar diurnal patterns, each having positive values in the morning and negative values in the afternoon that over the course of the diurnal cycle partially cancel the Twomey effect. Panel b shows the total ERF<sub>ACI</sub>, which is largely positive in the morning and approximately zero in the afternoon. Note that the sum of the three partial contributions from panel a, approximately calculated using Eqs. 7–9, agrees well with the total adjustment directly calculated from the LES as the difference in  $F^{\uparrow}$  between the purely polluted and pristine cases:

$$S = \frac{dF^{\uparrow}}{d \ln N_c} = \frac{F^{\uparrow}(N_{c200}, \text{LWP}_{c200}, f_{c200}) - F^{\uparrow}(N_{c25}, \text{LWP}_{c25}, f_{c25})}{\ln N_{c200} - \ln N_{c25}} \,. \tag{10}$$

Overall, the cloud adjustments  $S_{LWP}$  and  $S_f$  average to approximately zero over the diurnal cycle, enhancing the Twomey effect in the morning but nearly canceling it out in the afternoon. The daylight average values of the three terms are  $S_N = 17.0 \, \mathrm{W \, m^{-2}}$ ,  $S_{LWP} = -6.0 \, \mathrm{W \, m^{-2}}$ , and  $S_f = 5.6 \, \mathrm{W \, m^{-2}}$ .

Additionally, we calculated these terms for distinct pollution regimes using Eqs. 7–9, that is comparing  $N_{200}$  and  $N_{100}$  for the more polluted regime, and  $N_{25}$  and  $N_{50}$  for the more pristine regime, making use of the intermediate simulation results. The results are presented in Table 1. Notably, the  $S_N$  term is similar across background microphysical conditions, indicating the relative constancy of the Twomey effect. In contrast, the cloud adjustments are a net increase in albedo in the  $N_{50-25}$  experiments and a net decrease in the  $N_{200-100}$  experiments. This sign inversion is reminiscent of the 'inverted-V' dependance of LWP on Nc seen in satellite data (Gryspeerdt et al., 2022). However, in this specific case the inverted-V in LWP results from a modestly positive  $S_f$  term in the pristine conditions, where changes in droplet number concentration more strongly influence autoconversion and a much more strongly negative  $S_{LWP}$  term in the polluted conditions, where changes in droplet number concentration more strongly influence cloud top entrainment.

The third composite diurnal cycle shown in Fig. 3 (panel c illustrates the evolution of dCRE, which follows a similar pattern to the total susceptibility of shortwave outgoing radiation shown in panel b. The strongest effect occurs in the late morning hours, reaching as much as  $130-140~{\rm W\,m^{-2}}$ , with slightly negative values in the early afternoon and approaching zero by the end of the day. Both the evolution of dCRE and the susceptibility of  $F^{\uparrow}$  to aerosols exhibit strong day-to-day variability, with the Twomey effect dominating on day 1, and other adjustments becoming more prominent farther from the continent, where aerosol number concentrations decrease (Fig. 2~e, f).

# 3.3 Role of Key Physical Processes/ Key Controls

Why does the diurnal pattern in ERF<sub>ACI</sub> seen in Figure 3 emerge? To demonstrate the relevant mechanisms, Figure 4 presents a composite diurnal comparison of the cloudy boundary layer structure for the pristine and polluted scenarios. The polluted scenario produces less precipitation than the pristine scenario at all hours of the day (panel *a*). The polluted cloud entrains more efficiently and grows deeper than the pristine cloud over night (panel *b*). The polluted cloud is substantially more turbulent than the pristine cloud over night (panel *e*). Panel d shows that while the changes in cloud LWP affect the radiative heating of the

cloud layer, the afternoon differences in the shortwave warming are nearly exactly canceled by the differences in the longwave cooling. The resultant difference in radiative heating is primarily due to an overnight increases in longwave cooling of the polluted case. While both clouds are well coupled to the surface fluxes over night, the polluted cloud becomes substantially less coupled than the pristine cloud throughout the sunlit hours (panel c). The decoupling index is defined here as the ratio of the moisture flux at cloud base to that near the surface (van der Dussen et al., 2013), providing a measure of how much of the surface flux reaches the cloud layer.

Overall, a picture emerges of a polluted cloud that grows substantially faster over night than the pristine cloud with enhanced LWP due to precipitation suppression and a deeper cloud layer. However, this enhanced growth of the polluted cloud results in a deeper boundary layer that is more easily decoupled from the surface fluxes during the subsequent afternoon hours. These results explain the consistently positive (early morning) and negative (afternoon) sensitivities in cloud fraction and liquid water path seen in Figures 2 and 3.

In our experimental design, modifying  $N_c$  influences three model processes directly: radiative transfer, autoconversion, and cloud water sedimentation. We perform a series of additional experiments where we impose the polluted  $N_c$  on a particular process rate while all other processes see the pristine  $N_c$  to demonstrate the importance of that process on the evolution of the boundary layer and cloud microphyscial properties. A description of these experiments is provided in Table 2. We show the evolution of four quantities to demonstrate the influence of the various processes. The first two are cloud microphysical quantities: LWP and rain water path (RWP). The second two are related to the structure of the boundary layer: inversion height  $z_{inv}$  and decoupling index. Figure 5 shows the evolution of these four quantities for the various experiments. Several conclusions can be formed from these results:

- The influence of  $N_c$  on radiative transfer has a marginal effect on the evolution of the cloudy boundary layer. This is shown by the similarity between Exp. 3 and the pristine scenario (panels a-d).
- The autoconversion process has a positive and distinctly diurnal influence on the LWP sensitivity, while the cloud water sedimentation process has a smaller, negative, and relatively constant influence (panel e).
- Both autoconversion and, to a lesser extent, cloud water sedimentation affect the precipitation suppression mechanism (panel f). The latter process indirectly influences rainwater production by removing cloud liquid from the cloud top, thus limiting the efficiency of autoconversion.
- Both autoconversion and cloud water sedimentation influence the entrainment efficiency and growth of the boundary layer. Autoconversion has a larger effect than cloud water sedimentation, and the two processes interact in a super-linear manner to influence entrainment efficiency (panel g).
- Both autoconversion and cloud water sedimentation contribute to the decoupling of the cloud layer from the surface (panel h), which is consistent with the fact that both processes individually affect the cloud top entrainment rate.

A key summary of these conclusions is that the autoconversion and the cloud water sedimentation processes have some similar influences on the development of the cloudy boundary layer. The reason for this is that both processes remove liquid

from the cloud top entrainment zone, thereby slowing the rate of precipitation production, decreasing the efficiency of the entrainment, and slowing the decoupling of the boundary layer. This is closely related to the dynamics of the Entrainment Interfacial Layer (EIL; Haman et al., 2007; Kurowski et al., 2009), where the removal of liquid from the cloud top influences the structure of the EIL, leading to changes in boundary layer growth. We also see that these processes interact in a non-linear way, particularly in their influence on the LWP and the entrainment rate. Furthermore, the strong diurnal cycle in the sensitivity of cloud properties is a result primarily of the autoconversion process, whereas the cloud water sedimentation process operates over a longer time scale.

Finally, we comment on the relative role of removal of liquid water from the EIL and sub-cloud evaporative cooling on the evolution of the cloud layer through evaluation of an additional sensitivity study in which the drizzle evaporation process is turned off under pristine conditions (see supplemental material). In the no-evaporation experiment the surface buoyancy flux weakens due to reduced subcloud-layer cooling and a reduced ocean—atmosphere temperature contrast (Fig. S3). Despite this weaker surface forcing, cloud-layer turbulent kinetic energy tends to be higher at night (Fig. S6), suggesting that it is not directly controlled by surface buoyancy flux but is instead primarily driven by longwave radiative cooling (Wood, 2012). Nonetheless, the entrainment rate is also reduced (Fig. S5) due to lower moisture availability in the cloud layer (Fig. S4). A similar reduction in surface buoyancy flux is observed in the polluted case, although the entrainment rate increases significantly at night because of greater moisture availability compared to the pristine case. Stevens et al. (1998) found that drizzle evaporation stabilizes the subcloud layer and reduces entrainment via decoupling. Uchida et al. (2010) noted that drizzle evaporation below cloud base dampens buoyancy flux, weakens turbulence, and reduces entrainment. However, this dynamical argument is not fully supported by our results as we find that rain evaporation is associated with increased cloud-top entrainment rates. Furthermore, during the day, both cloud-layer turbulence and entrainment decrease more strongly with active evaporation than without evaporation, highlighting a pronounced diurnal modulation in the scenarios analyzed in this study.

## 4 Conclusions

This paper analyzes the diurnal susceptibility of shallow subtropical clouds to aerosols using a six-day Lagrangian LES along the stratocumulus-to-cumulus transition with realistic environmental forcing including a diurnal cycle of solar radiation. Pristine and polluted scenarios are simulated to quantify the ERF<sub>ACI</sub> and its component terms. The ERF<sub>ACI</sub> is broken down into the Twomey effect, a LWP<sub>c</sub> adjustment, and a  $f_c$  adjustment. The daytime average values of the three terms are approximately 17 W m<sup>-2</sup>, -6 W m<sup>-2</sup>, and 6 W m<sup>-2</sup>, respectively. However, there is a substantial diurnal cycle in the three terms. The 305 Twomey effect is always positive and most efficient in the morning hours because  $f_c$  is larger in the morning than in the afternoon, although it also has a significant positive contribution during the afternoon. More importantly, the LWP<sub>c</sub> and  $f_c$  adjustments switch signs from positive in the morning to negative in the afternoon. The resulting diurnal pattern of the ERF<sub>ACI</sub> is super-Twomey in the morning and near neutral in the afternoon. Results further show evidence of a sign inversion (inverted-V) in the cloud adjustment terms with positive cloud adjustments in pristine conditions and negative cloud adjustments in polluted conditions.

The reason this diurnal pattern in ERF<sub>ACI</sub> emerges is that the diurnal amplitude of the cloud extent is increased relative to the pristine case. This occurs because precipitation is suppressed in the polluted cloud relative to the pristine resulting in a thicker more turbulent cloud layer, with enhanced longwave cooling and increased cloud liquid water near the cloud top entrainment zone during the nighttime hours. As a result of the increase cloud top liquid water, the polluted cloud entrains more efficiently and grows substantially faster and deeper overnight. However, this nighttime success of the polluted cloud is not sustainable as it results in a boundary layer that is deeper, drier and more decoupled, which ultimately leads to a stronger mid-day collapse of the cloudy boundary layer the following afternoon.

A key mechanism in the causal chain is the increase in cloud top liquid water with increases in  $N_c$ . Through sensitivity experiments it is shown that both sedimentation of cloud and rain water are effective at reducing the efficiency of the entrainment. However, cloud sedimentation and autoconversion interact in a nonlinear manner to result in a combined effect on entrainment that is greater than the sum of each term. This could occur due to the non-linearity of the autoconversion process interacting with a reduced amount of cloud liquid water at cloud top due to the cloud water sedimentation. Therefore, accurate simulation of the entrainment drying mechanism in global models should include both cloud and rain water sedimentation as is the case in at least one commonly used cloud microphysics parameterization (Morrison and Gettelman, 2008).

The findings of this study are in qualitative agreement with a growing body of literature based on both modeling and observations that increasing  $N_c$  causes an amplification of the diurnal cycle of cloud properties which subsequently causes a morning/afternoon contrast in the sign of the cloud adjustments with adjustments enhancing Twomey brightening in the morning and offsetting the brightening in the afternoon. The result in this study is a negligible diurnal average effect of the adjustments on the ERF<sub>ACI</sub>. However, this is based on a single suite of simulations and we must be cautious in extrapolating these results to more general conditions. In particular, a key mechanism that mediates the diurnal response in these simulations is the suppression of precipitation. We have no expectations that increasing in  $N_c$  in non-precipitating clouds would have the same effect on the diurnal cycle. We could speculate that in that case the cloud adjustments would be robustly negative across the diurnal cycle. Indeed our limited simulations here demonstrate that the cloud adjustments change sign from positive to negative as  $N_c$  is increased. Future research is necessary to extend the Lagrangian approach used here to many more trajectories representative of the diversity of atmospheric conditions to fully understand the influence of the diurnal cycle of the cloud adjustments on the ERF<sub>ACI</sub>.

- . The System for Atmospheric Modeling code is available upon contacting Dr. Marat Khairoutdinov.
- . Trajectory data, model inputs and outputs needed to reproduce the figures are available at: https://zenodo.org/records/14873449

## Appendix A

340 The cloud albedo is calculated using the hybrid model of Meador and Weaver (1980), which includes a dependence on the solar zenith angle

$$\alpha_{cld} = \frac{1}{1 + \gamma_1 \tau_c} \left( \gamma_1 \tau_c + (\beta_o - \gamma_1 \mu_o) \left( 1 - \exp\left(\frac{-\tau_c}{\mu_0}\right) \right) \right) . \tag{A1}$$

The two  $\gamma$  coefficients of this model are given by

$$\gamma_1 = \frac{7 - 3g^2 - \omega_o(4 + 3g) + \omega_o g^2(4\beta_o + 3g)}{4(1 - g^2(1 - \mu_0))} \tag{A2}$$

and

$$\gamma_2 = \frac{1 - g^2 - \omega_o(4 + 3g) - \omega_o g^2(4\beta_o + 3g - 4)}{-4(1 - g^2(1 - \mu_0))} , \tag{A3}$$

where  $\omega_o$  is the single scatter albedo and g is the asymetery parameter. The third coefficient is given by

$$\beta_o = \frac{1}{2\omega_*} \int_0^1 P(\mu_o, -\mu') \, d\mu' \quad , \tag{A4}$$

which is the fraction of single scattered radiation out of the solar beam into the backscattering hemisphere. The single scattering phase function (P) is subject to the normalization condition

$$\frac{1}{4\pi} \int_{-1}^{1} \int_{0}^{2\pi} P(\mu, \phi; \mu', \phi') d\phi' d\mu' = \omega_o.$$
 (A5)

The inclusion of the  $\beta_o$  term is a complication as in general it represents an integral that can not be represented analytically. In this work, we parameterize this integral based on numerical integration of the Henyey and Greenstein (1941) phase function for g = 0.86 and  $\omega_o = 1$  giving the follow approximate formulation

$$\beta_o(g = 0.86, \omega_o = 1) \approx 0.5 \exp{-2.7\mu_o^{0.7}}$$
, (A6)

where the Henyey-Greenstein phase function subject to the proper normalization is given by

$$P_{HG} = \omega_o \frac{1 - g^2}{(1 + g^2 - 2g\cos(\Theta))^{\frac{3}{2}}} . \tag{A7}$$

- . MK and ML designed the experiments and MK carried them out. KS prepared the Lagrangian trajectory and observational data. MK developed the model code and performed the simulations. MK and ML preformed the analysis. MK prepared the manuscript with contributions from all co-authors.
  - . At least one of the (co-)authors is a member of the editorial board of Atmospheric Chemistry and Physics.

. The research was carried out at the Jet Propulsion Laboratory, California Institute of Technology, under a contract with the National Aeronautics and Space Administration (80NM0018D0004). Texas Advanced Computing Center (TACC) The University of Texas at Austin is acknowledged for providing high-performance computing resources. Work from LLNL is performed under the auspices of the US DOE by LLNL under Contract DE-AC52-07NA27344: IM Number LLNL-ABS-867332. LLNL-JRNL-2002664.

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

**Table 1.** Composite daytime sensitivity of upward radiative flux to  $\ln N_c$  under different aerosol regimes. Note that the full-range  $S_N$  is smaller than the pristine and polluted ones, which is due to a stronger stochastic variability between the  $N_{200}$  and  $N_{25}$  solutions.

|                                   | Pristine                      | Polluted                      | Full range                    |
|-----------------------------------|-------------------------------|-------------------------------|-------------------------------|
| $\frac{dF^{\uparrow}}{d \ln N_c}$ | $(N_{50}-N_{25})$             | $(N_{200} – N_{100})$         | $(N_{200} – N_{25})$          |
|                                   | $(\mathrm{W}\mathrm{m}^{-2})$ | $(\mathrm{W}\mathrm{m}^{-2})$ | $(\mathrm{W}\mathrm{m}^{-2})$ |
| $S_N$                             | 21.3                          | 22.8                          | 17.0                          |
| $S_{\rm LWP}$                     | -0.1                          | -13.0                         | -6.0                          |
| $S_f$                             | 11.3                          | 3.2                           | 5.6                           |

**Table 2.** Summary of sensitivity experiments with varied microphysical processes. N25 refers to droplet number concentration for the pristine case, whereas N200 for the polluted case. Note that all the modifications in the Ex3-Ex5 experiments relate to the baseline pristine case.

| Experiment | Radiation | Autoconversion | Sedimentation | Description                           |
|------------|-----------|----------------|---------------|---------------------------------------|
| Ex1        | N200      | N200           | N200          | Polluted case                         |
| Ex2        | N25       | N25            | N25           | Pristine case                         |
| Ex3        | N200      | N25            | N25           | Impact of pollution on radiation      |
| Ex4        | N25       | N200           | N25           | Impact of pollution on autoconversion |
| Ex5        | N25       | N25            | N200          | Impact of pollution on sedimentation  |

**Figure 1.** Overview of the analyzed case: (a) Lagrangian trajectory, and the evolution of (b) liquid water path from LES and observations, (c) cloud top height from LES and observations, (d) curtain plot of cloud water and rain water mixing ratios for the polluted case (N200), (e) observed and prescribed in LES droplet number concentrations, (f) sea surface temperature (SST), (g) observed and simulated cloud fraction, (h) cloud optical thickness from the LESs, (i) curtain plot of cloud water and rain water mixing ratios for the pristine case (N25).

Figure 2. Six diurnal cycles of (a) cloud liquid water path  $(LWP_c)$  and its difference between the polluted and pristine cases, (b) cloud fraction and its difference between the polluted and pristine cases, (c) cloud albedo calculated following Meador and Weaver (1980) and its difference between the polluted and pristine cases, (d) incoming solar shortwave energy flux, (e) cloud radiative effect, and its difference between the polluted and pristine cases, and (f) the susceptibility of shortwave outgoing radiation to droplet number concentration decomposed into the three parts: Twomey effect (Eq. 7), LWP adjustment (Eq. 8), and cloud fraction adjustment (Eq. 9).

Figure 3. Composite diurnal cycle of: (a) the susceptibility terms  $S_N$ ,  $S_{LWP}$ ,  $S_f$  from Eqs. 7-9, calculated offline using the differences between the N200 and N25 simulations, and (b) their sum (magenta) compared against the actual LES model output (black) from Eq. 10. Panel (c) shows the diurnal cycle of dCRE.

**Figure 4.** Composite diurnal cycles for the pristine (red) and polluted (blue) scenarios, showing (a) LWP and RWP, (b) cloud-top entrainment rate, (c) PBL decoupling index, (d) cloud-layer shortwave (SW) and longwave (LW) radiative tendencies and their difference between the two scenarios (magenta), and (e) cloud-layer turbulent kinetic energy (TKE<sub>c</sub>). Text in blue highlights features specific to the polluted case.

**Figure 5.** Results of sensitivity experiments for two extreme droplet number concentrations, polluted (200 cm<sup>-3</sup>) and pristine (25 cm<sup>-3</sup>), applied independently to three main model components: radiation (RAD), rain autoconversion (AUT), and cloud water sedimentation (SED), as explained in Tab. 2. The panels show time series of: (a) LWP and (e) its difference between three pairs of key experiments (Ex1–Ex2, Ex4–Ex2, Ex5–Ex2); (b) RWP and (f) its difference; (c) inversion height and (g) its difference; (d) PBL decoupling index and (h) its difference (with nighttime values omitted for clarity since the PBLs are coupled in all cases). To reduce noise and extract the main signal, the LWP/RWP and decoupling index time series are smoothed using a 5-hour and an 8-hour window, respectively.