# Peer review of "The Diurnal Susceptibility of Subtropical Clouds to Aerosols"

_EGUsphere, 2025_

## Referee Comment (RC1)

**Review of "The Diurnal Susceptibility of Subtropical Clouds to Aerosols" by Kurowski et al. (egusphere-2025-714)**

This study analyzes how changes in the aerosol influence cloud properties, focusing on how the diurnal cycle affects the cloud response. For this, the authors analyze the development of a maritime boundary layer, following it from the subtropical Southeastern Pacific towards the equator for six days. The authors show that polluted clouds are brighter than pristine clouds during the night and morning. The associated buildup of cloud water results in stronger precipitation during the subsequent hours of the day, which causes the dissipation of polluted clouds in the afternoon. The authors combine high-resolution large-eddy simulations (LESs) with observations for this study. Although these findings are of interest to the current discussion on aerosol-cloud interactions (ACIs), the manuscript requires major revisions before I can support its publication.

**Major Comments**

*Writing.* Although I can follow the manuscript and understand its main ideas, the authors should invest some time to improve the presentation of their results. The introduction (and, to a lesser degree, the abstract) misses the opportunity to state the central question addressed in this manuscript. Equations are a part of a sentence and require adequate punctuation. Abbreviations, once introduced, should be used throughout the manuscript and not re-introduced in the middle of the manuscript (e.g., sea surface temperature). Figure captions should describe what is shown in the corresponding figure (Fig. 2).

*A deeper discussion of previous literature.* A deeper discussion of previous literature on this subject is required in some places. While the study of Prabhakaran et al. (2024) is mentioned, the authors do not discuss the increased absorption of shortwave radiation of polluted clouds during the daytime, which Prabhakaran et al. (2024) identified as a major reason leading to the dissipation of polluted clouds in the afternoon. While it is plausible that more substantial precipitation also causes this dissipation, as shown by the authors, the authors should try to compare the different influences of these processes more directly.

*Composite diurnal cycle.* The boundary conditions along the investigated trajectory change over six days. However, the authors create composite diurnal cycles (ll. 176 – 184, Fig. 3) to analyze changes in the susceptibility from all six days in a single panel. Are changes in the boundary conditions negligible for the susceptibility? What is the day-to-day variability in the susceptibility?

**Minor Comments**

Ll. 11 – 12: Please clarify what you mean by "This entrainment enhancement is mediated by the sedimentation of cloud and precipitation water from the entrainment zone."

L. 25: One might define the liquid water path as the vertically integrated liquid water content.

L. 32: $\tau_c$ is not defined. Why is $N_c$ not used here?

Ll. 34 – 44: One should emphasize that cloud water adjustments (and probably cloud fraction adjustments mentioned below) are a function of $N_c$, which enables the coexistence of positive and negative adjustments without contradiction.

Ll. 79 – 80: It is unclear whether "methodology " refers to "selection" or "production".

Ll. 100 – 101: Please clarify "sharpening the inversion layer to around 40 m".

Ll. 105 – 106: What is weakly homogenized?

Ll. 106 – 108: Khairoutdinov and Kogan (2000) is a two-moment cloud microphysics scheme. Was it applied as a one-moment scheme? How did the authors do that?

Ll. 108 – 109: Please elaborate on the evolution of the droplet number concentration. What is the initial value? Is the decrease/increase in $N_c$ prescribed or predicted by the cloud microphysics scheme?

L. 124: Add a reference for the treatment of "multiple reflections".

Eq. 5: Is $A_{cld}$ different from $A_c$?

Ll. 137 – 139: Does this sentence indicate that the overbar denotes a temporal average over the entire six days of simulation? How did the authors ensure that only one of the parameters ($N_c$, $LWP_c$, or $f_c$) is varied for calculating the susceptibilities (7) to (9)? I believe that this is almost impossible to achieve in an interactive simulation.

Ll. 140 – 142: How does the sentence "the results are intended […]" relate to the part before the comma?

L. 159: State explicitly that only a case corresponding to the pristine scenario has been observed.

L. 174: Why is the cloud albedo approaching unity? The clouds assessed here are relatively shallow, with a cloud albedo substantially smaller than unity.

Ll. 174 – 175: Please refer to a figure to substantiate these claims.

Fig. 3: What causes the disagreement between the black and pink line?

L. 184: How do these values compare to other values presented in the literature?

Ll. 194 – 195: What "near-surface flux" needs to be considered?

Ll. 205 – 208: Explain what is changed in the different experiments, and to what experiment 5 refers to.

Fig. 5h: Why are nights omitted? There is also strong entrainment during the nights which could cause decoupling.

Ll. 217 – 218: Decoupling is not only caused by increased entrainment. Evaporation of drizzle below the cloud base is another important factor to consider.

Ll. 233 – 234: Based on the explanations on the susceptibilities (7) to (9), I was assuming that the Twomey effect is calculated using a constant cloud fraction.

**Technical Comments**

L. 42: Change reference style.

L. 85: Define PBL.

L. 109: Use $cm^{-3}$ instead of $cc^{-1}$.

Fig. 5: Change "subsidence" to "sedimentation".

---

## Author Comment (AC1)

Thank you to both reviewers for their insightful comments. We have made major changes to the manuscript to address the reviewers' comments. Most significantly, we have:

- analyzed additional experiments with Nd = 50/cc and Nd = 100/cc to identify that our results do show signs of the inverted-V LWP response,
- performed an additional sensitivity experiment in which evaporation of precipitation is disabled to evaluate the sensitivity of the subcloud moistening/cooling on the cloud evolution,
- strengthened the referencing in the introduction.

Our point-by-point responses to all of the referee's comments are below. The referee comments are in black, and our responses are in blue text.

**Response to Reviewer #1**

Major Comments

Writing. Although I can follow the manuscript and understand its main ideas, the authors should invest some time to improve the presentation of their results. The introduction (and, to a lesser degree, the abstract) misses the opportunity to state the central question addressed in this manuscript. Equations are a part of a sentence and require adequate punctuation. Abbreviations, once introduced, should be used throughout the manuscript and not re-introduced in the middle of the manuscript (e.g., sea surface temperature). Figure captions should describe what is shown in the corresponding figure (Fig. 2).

Thank you for this comment. We carefully reviewed the text and made the necessary changes. This includes clarifying the abstract and the introduction, fixing the punctuation, abbreviations, and improving figure captions. In particular, note lines 90-97 which state the central question in the introduction.

One major change, made with the potential reader in mind, is that we decided to switch from dlnF/dlnN to dF/dlnN. This change is reflected in the equations and figures, and the susceptibility is now expressed in units of $W/m^2$. We believe that presenting the results in terms of $W/m^2$ will make the results more approachable to the average reader.

A deeper discussion of previous literature. A deeper discussion of previous literature on this subject is required in some places.

Thank you for pointing this out. We deepened the discussion in the introduction, adding more references where appropriate.

While the study of Prabhakaran et al. (2024) is mentioned, the authors do not discuss the increased absorption of shortwave radiation of polluted clouds during the daytime, which Prabhakaran et al. (2024) identified as a major reason leading to the dissipation of polluted

clouds in the afternoon. While it is plausible that more substantial precipitation also causes this dissipation, as shown by the authors, the authors should try to compare the different influences of these processes more directly.

We believe that our results are, to a large extent, consistent with those of Prabahakaran et al (2024). Consistent with most of their results, our study directly shows that suppressed precipitation in the polluted scenario drives stronger nighttime entrainment, which in turn leads to enhanced daytime decoupling and a more rapid afternoon LWP collapse. However, we do come to a different conclusion regarding the role of SW absorption in causing the afternoon cloud dissipation in the polluted scenario. We show the diurnal cycle of both SW and LW cloud-layer heating / cooling in Figure 4d. Here we show some similarities and differences with Prabahakaran et al. (2024). Like Prabahakaran et al. (2024) we do see an increase in SW absorption in the polluted scenario around 10h in the morning. However, we see that throughout the majority of the day the SW absorption is reduced in the polluted scenario owing to the reduced LWP. More consequentially, we show that any perturbation in the SW absorption is canceled by a nearly equal (in fact slightly stronger) perturbation in the LW cooling. As a result we see that the net radiative perturbation throughout the day is in fact a cooling in the polluted scenario (magenta curve in Figure 4d.). The differences between our conclusions and Prabahakaran may well be due to the different meteorological conditions simulated. Parbahakaran's Figure 4 only shows SW absorption and not the net heating so we cannot assess whether a similar cancelation may be occurring in their simulations. We really require future study with a more comprehensive set of trajectories to more comprehensively separate the roles of SW and LW radiation in determining the cloud response in microphysically perturbed clouds.

Composite diurnal cycle. The boundary conditions along the investigated trajectory change over six days. However, the authors create composite diurnal cycles (ll. 176 – 184, Fig. 3) to analyze changes in the susceptibility from all six days in a single panel. Are changes in the boundary conditions negligible for the susceptibility? What is the day-to-day variability in the susceptibility?

The purpose of the composite diurnal cycle is to synthesize results and help generalize the conclusions in the paper. Since all simulations begin with polluted air masses off the continental coast (Fig. 1; based on observations, as explained in the text), the first day shows small differences across cases, with variations developing over the following days. While we simulate the subtropical transition from stratocumulus to cumulus, which naturally involves day-to-day differences, capturing that variability is not the primary focus of this study. Instead, the composite diurnal cycle is used to filter out such variations and highlight the overall diurnal susceptibility of the cloud layers, considering all relevant physical interactions under an observation-based scenario. Please note that our approach helps avoid complications related with a spin up time as well.

For the boundary conditions, we assume the reviewer asks about the latent and sensible heat fluxes at the surface, since the SST evolution as a boundary condition remains the same for all the simulations. Plots of the surface heat fluxes have been added to

supplemental material. The evolution of the fluxes indeed changes and for drier (i.e., less precipitating) PBLs, they tend to decrease by around 8-10 W/m2 for sensible heat fluxes and increase by comparable or somewhat larger absolute values for latent heat fluxes, with a strong diurnal cycle. Note that in terms of water supply, the fluxes change maximally by 8-10 %. We now discuss it in the paper.

Day-to-day susceptibility is now plotted in Fig. 2 (f). Except for day 1, the differences remain comparable throughout the transition. Our Figs. 2 and 3 also shows how the diurnal evolution of cloud and radiative properties, including CRE and dCRE look like – they are similar for each of the days 2-6, although the impact of transitioning to shallow Cu can also be seen.

Minor Comments

Ll. 11 – 12: Please clarify what you mean by "This entrainment enhancement is mediated by the sedimentation of cloud and precipitation water from the entrainment zone."

This was not worded very well. We've changed the sentence to be more direct.

L. 25: One might define the liquid water path as the vertically integrated liquid water content.

LWP is widely understood as the vertical integral of lwc. We will leave this text as it was.

L. 32: $\tau_c$ is not defined. Why is $N_c$ not used here?

Thanks. We have added the definition: *cloud optical depth (line 24)*

Ll. 34 – 44: One should emphasize that cloud water adjustments (and probably cloud fraction adjustments mentioned below) are a function of $N_c$, which enables the coexistence of positive and negative adjustments without contradiction.

Great point. We have added the following text:

We note that cloud adjustments are dependent on the background $N_c$, which can explain the presence of both positive and negative adjustments without contradiction. This state dependence in the cloud adjustment is manifest as the 'inverted V' relationship between $N_c$ and $LWP_c$ implying postive adjustment at low $N_c$ and negative adjustment at high $N_c$ (Gryspeerdt et al., 2022).

Reference:

Gryspeerdt, E., Glassmeier, F., Feingold, G., Hoffmann, F., and Murray-Watson, R. J.: Observing short-timescale cloud development to constrain aerosol–cloud interactions, Atmos. Chem. Phys., 22, 11727–11738, https://doi.org/10.5194/acp-22-11727-2022, 2022.

Ll. 79 – 80: It is unclear whether "methodology " refers to "selection" or "production".

We rephrased that sentence to clarify that the methodology refers to the way the trajectories were produced (Lines 99-123).

Ll. 100 – 101: Please clarify "sharpening the inversion layer to around 40 m".

We've edited the sentence to clarify the message. In general, MERRA-2 data are too smooth to provide reliable information about inversion strength and using them directly in LES fails to reproduce stratocumulus. Past field campaigns and modeling studies have shown that stratocumulus forms under a much stronger and sharper inversion. To reconstruct the stratocumulus layer in our simulations, we sharpened the inversion by reducing its thickness to around 40m while keeping the same temperature and water vapor mixing ratio differences.

Ll. 105 – 106: What is weakly homogenized?

Temperature and water vapor mixing ratio fields are weakly homogenized. We now explain that in the text. Based on our experience, LES models that use periodic boundary conditions and are run for long durations on large domains tend to develop spurious internal circulations that depend on the domain size. For instance, if one region becomes dominated by convective upwelling, another region may develop compensating subsiding motion. This mechanism is suppressed here by introducing weak homogenization of the temperature and moisture fields. Arguably, this adjustment does not affect the smaller scales of motion associated with convection and turbulence.

Ll. 106 – 108: Khairoutdinov and Kogan (2000) is a two-moment cloud microphysics scheme. Was it applied as a one-moment scheme? How did the authors do that?

Yes, indeed. Thank you for that comment. As explained in the text (Lines 130-140) and showed in Fig. 1e, we prescribe cloud droplet number concentrations using observations and perturbing the observations to represent the polluted cases away from the continent as well. This way, we are also able to calculate dlnNd in a straightforward manner.

Ll. 108 – 109: Please elaborate on the evolution of the droplet number concentration. What is the initial value? Is the decrease/increase in Nc prescribed or predicteed by the cloud microphysics scheme?

As shown in Fig. 1e, the initial droplet number concentrations range between 450 and 600/cc, representing the polluted region near the continental coast. This evolution is prescribed as stated in lines 130-140. All droplet number concentrations are prescribed to rapidly decrease to their asymptotic values during the first day, following observations. This ensures that each case starts with a non-precipitating stratocumulus layer, which begins to precipitate later, helping to maintain consistency and stability in the initial state after model spin-up.

L. 124: Add a reference for the treatment of "multiple reflections".

We have added the Stephens et al., (1984) reference:

Stephens, G. L., 1984: The parametrization of radiation for numerical weather prediction and climate models. *Mon. Wea. Rev.*, **112**, 826–866, doi:10.1175/1520-0493(1984)112<0826:TPORFN>2.0.CO;2.

Eq. 5: Is Acld different from Ac?

This was a typo. $A_{cld}$ and $A_c$ represented the same quantity. Now $A_{cld}$ is changed to $A_c$.

Ll. 137 – 139: Does this sentence indicate that the overbar denotes a temporal average over the entire six days of simulation? How did the authors ensure that only one of the parameters (Nc, LWPc, or fc) is varied for calculating the susceptibilities (7) to (9)? I believe that this is almost impossible to achieve in an interactive simulation.

The overbar just means that the pristine and polluted values are both used to calculate the average, and that value is a function of time. We have clarified the text on line 170.

Ll. 140 – 142: How does the sentence "the results are intended [...]" relate to the part before the comma?

That paragraph has been removed as we now show dF/dlnN.

L. 159: State explicitly that only a case corresponding to the pristine scenario has been observed.

We have added this sentence:

*Note that the pristine scenario best matches the observations and the polluted scenario should be interpreted as a perturbation from the observed state.*

L. 174: Why is the cloud albedo approaching unity? The clouds assessed here are relatively shallow, with a cloud albedo substantially smaller than unity.

We think the reviewer may have misinterpreted this sentence. We are not suggesting that the clouds here have albedo near unity but rather the general tendency that as albedo increases towards unity the Twomey effect decreases towards zero.

Ll. 174 – 175: Please refer to a figure to substantiate these claims. Fig. 3: What causes the disagreement between the black and pink line?

Added reference. We also explain now the differences between the two lines and provide one more equation to clarify that:

"(see Figure 2d/e)

Note that the sum of the three partial contributions from panel $a$, approximately calculated using Eqs.~7–9, agrees well with the total adjustment directly calculated from the LES as the difference in $F^{\uparrow}$ between the purely polluted and pristine cases:"

The difference between the black and pink lines in Fig. 3b arises because each S_x term shown in panel (a) of Fig. 3 was computed using Eqs. 7–9, with various input terms for F

taken as the average between the pristine and polluted cases. To our knowledge, there is no direct way to compute these individual terms from the LES, so our decomposition is necessarily approximate. However, the LES does allow for a direct calculation of the total adjustment simply defined as the difference in F between the two simulations. Thus, the final comparison also serves as a sanity check, demonstrating that our offline decomposition method produces results that closely match those from the LES.

L. 184: How do these values compare to other values presented in the literature?

This is an important point. However, we find it difficult to compare our results to other papers since our case is distinct. Although there are papers looking at this effect, their conditions are different from the ones we simulate (regions, seasons, atmospheric state) and thus we prefer to avoid comparing direct numbers to avoid confusion. We would appreciate any suggestions on potential papers we could use for such a comparison that we may have overlooked.

Ll. 194 – 195: What "near-surface flux" needs to be considered?

Thank you for noticing that. It is the moisture flux, which we now clarify in the text (lines 247-249)

Ll. 205 – 208: Explain what is changed in the different experiments, and to what experiment 5 refers to.

All the experiments are explained in the legend of Fig. 5, and in Table 2. Experiment 5 refers to the case in which radiation (RAD) and autoconversion (AUT) use pristine conditions (i.e., apply droplet number concentration of N25), whereas cloud water subsidence applies polluted conditions. So the difference between Ex2 and Ex5 is only in the way terminal velocity is calculated for cloud water.

Fig. 5h: Why are nights omitted? There is also strong entrainment during the nights which could cause decoupling.

We now clarify in the text that nights are omitted for clarity. Time series of the decoupling index are still shown in Fig. 5d. However, because all of the cloud layers are coupled, differences between them are irrelevant yet would generate a more busy figure. What matters is the level of decoupling during daytime.

Ll. 217 – 218: Decoupling is not only caused by increased entrainment. Evaporation of drizzle below the cloud base is another important factor to consider.

Great point. We discuss that in the new last paragraph added to the results section (lines 284-297), as supported by more results showed in the supplemental material.

Ll. 233 – 234: Based on the explanations on the susceptibilities (7) to (9), I was assuming that the Twomey effect is calculated using a constant cloud fraction.

We have updated the equations to better explain how we calculate those terms. In this case, "constant" means the same value for both F values in the numerator at any time

(although those values change over time. When calculating F=F(Nc,LWPc,fc) in eqs (7-9), there is always one quantity that changes {Nc,LWPc,fc}, whereas the remaining two are kept "constant", which means they still evolve in time, but always represent the mean value between the pristine and polluted ones.

Technical Comments

L. 42: Change reference style.

Corrected.

L. 85: Define PBL.

Done.

L. 109: Use cm-3 instead of cc-1 .

Done.

Fig. 5: Change "subsidence" to "sedimentation".

Done.

---

## Author Comment (AC2)

Thank you to both reviewers for their insightful comments. We have made major changes to the manuscript to address the reviewers' comments. Most significantly, we have:

- analyzed additional experiments with Nd = 50/cc and Nd = 100/cc to identify that our results do show signs of the inverted-V LWP response,
- performed an additional sensitivity experiment in which evaporation of precipitation is disabled to evaluate the sensitivity of the subcloud moistening/cooling on the cloud evolution,
- strengthened the referencing in the introduction.

Our point-by-point responses to all of the referee's comments are below. The referee comments are in black, and our responses are in blue text.

**Response to Reviewer #2**

MAJOR COMMENTS:

The discussion of the physical processes influencing the results is somewhat underdeveloped. For instance, in Figure 4, entrainment, coupling, and radiative heating are shown in different units, which makes direct comparisons between them challenging. While these processes can be qualitatively associated with the evolution of LWP, the relative importance of each remains unclear, weakening the overall argument. The role of precipitation is neglected. In particular, a relevant mechanism to consider is the evaporation of drizzle below the cloud base, which dampens buoyancy flux, weakens turbulence, and reduces entrainment. See the Introduction section in Uchida et al. (2010, doi:10.5194/acp-10-4097-2010) for a summary of this mechanism. How does this process influence the results presented in this study?

Than you for this comment. While we find it to be a very important point, it is not so obvious how to assess the impact of different processes in a credible way using the same units, because they can modify the system in a different way, on different time scales, and those non-linear interactions can accumulate over long times differently. Also, the way we present our results is one of possible approaches, used in many previous studies (e.g., Stevens et al. 2005, Sandu and Stevens 2011, van der Dussen et al. 2013, Bretherton and Blossey 2014, Chun et al. 2023, Chun et al. 2025).

However, to make the comparison more quantitative in terms of state variables, we have chosen an alternative approach, in which we run a set of sensitivity experiments accounting for all non-linear interactions, while adjusting a single process controlling the development of the system (see Tab 2 in the revised paper). Noteworthy, our set of

sensitivity experiments includes the runs with precipitation strongly suppressed, as well as with the evaporation of precipitation disabled (the latter in the supplemental material). These tests help to better quantify the very impact of precipitation on the transition.

One major change, made with the potential reader in mind, is that we decided to switch from dlnF/dlnN to dF/dlnN. This change is reflected in the equations and figures, and the susceptibility is now expressed in units of W/m$^2$. We believe that presenting the results in terms of W/m$^2$ will make the results more approachable to the average reader.

As the authors noted, there have been many papers on the ACI in marine shallow clouds. It would greatly benefit the readers and strengthen and current paper if the authors can more clearly connect their new findings to the existing body of work. This is not about advertising any previous studiy, but about highlighting the novel aspects of the current paper through thoughtful comparison. For example, the diurnal LWP pattern, higher at night and lower during the day, is reminiscent of the results in Sandu et al. (2008), who proposed an explanation for this behavior. While a comprehensive literature review or exhaustive testing of all previous hypotheses is not necessary, a deeper and more explicit discussion of how this study builds upon or diverges from past work would be highly valuable.

In the introduction, we cite several previous studies that we are aware of that specifically address the diurnal cycle of the sensitivity of shallow clouds to aerosol perturbations. For example, the Sandu et al. (2008) study referenced by the reviewer was already cited on (now) line 75. We also added more citations to the text.

MINOR COMMENTS:

I found it concerning that some well-established ideas are cited using only recent publications, rather than the original sources. For example, citing Wall et al. for ERFaci and Hoffmann et al. (2023) for Eq. 6. Even a quick check of the cited papers would tell references to the original sources, which the authors should cite directly to properly acknowledge the historical development of these concepts.

Thanks. We have added a couple of the foundational references where appropriate. On line 20, we add reference to the AR5 IPCC report (Boucher et al., 2013) where the concept of ERF as opposed to RF was first socialized on a large scale across multiple forcings including for ACI and CO2.

Reference: Boucher, O. et al., 2013: Clouds and Aerosols. In: Climate Change 2013: The Physical Science Basis. Contribution of Working Group I to the Fifth Assessment Report of the Intergovernmental Panel on Climate Change [Stocker, T.F., D. Qin, G.-K. Plattner, M. Tignor, S.K. Allen, J. Boschung, A. Nauels, Y. Xia, V. Bex, and P.M. Midgley (eds.)]. Cambridge University Press, Cambridge, United Kingdom and New York, NY, USA, pp. 571–657, doi:10.1017/cbo9781107415324.016.

We add a classic reference (Brenguier et al., 2000) for the adiabatic cloud optical depth calculation but keep the Hoffman et al reference since there is a parameter choice which is needed and we use the specific parameter given by Hoffman. The text is changed as follows:

*'The cloud optical depth is calculated at each time step from the domain mean time-dependent modeled LWPc and Nc assuming an adiabatic cloud vertical structure (Brenguier et al., 2000) following the specific implementation of Hoffmann et al. (2023).*

Reference: Brenguier, J., H. Pawlowska, L. Schüller, R. Preusker, J. Fischer, and Y. Fouquart, 2000: Radiative Properties of Boundary Layer Clouds: Droplet Effective Radius versus Number Concentration. *J. Atmos. Sci.*, **57**, 803–821, https://doi.org/10.1175/1520-0469(2000)057<0803:RPOBLC>2.0.CO;2.

Please clarify the connection between Eqs. 7-9 and Eq. 1. Are these equations intended as proxies for specific terms in Eq. 1? Please state clearly.

Here we refer the reviewer back to Equation 1 where the three terms ($S_N$), $S_{LWP}$, and $S_f$ are denoted underneath the curly brackets. We added a clarification at line 164 that we are referring to the terms in equation 1:

*'The offline radiation calculations are used to decompose the ERFACI into the three indirect sensitivity terms defined in Eq. 1.'*

We also corrected two problems with eqs. 7-9. First, we now write them in the same form as equation 1 and second, we also add an approximately equal sign to account for the fact that we are estimating the derivatives.

Line 149: Why does the current work focus only on two aerosol scenarios? More specifically, did the authors observe any indication of the previously reported "inverted-V" shape in the LWP-N relationship? If not, it is worth noting. This comment is not meant to cast doubt on the results, but rather to encourage a more complete discussion of their implications.

Thanks for this comment. We do see some evidence for the inverted-V phenomenon. We now comment on the "inverted-V" shape in the text (lines 48-50, 228-232-250, 308-310). We also included more results based on the N50 and N100 results as well. They are summarized in Tab. 1, where the susceptibility of different terms is calculated for the more polluted (N200-N100) and more pristine (N50-N50) sides.

Line 180: The term "similar" seems too vague here. Please be more specific about the particular features the authors intended to highlight.

Good point. This was vague. We have rewritten the sentence as follows (lines 214-216):

*The other two cloud adjustment terms ($S_{LWP}$, $S_f$) have similar diurnal patterns, each having positive values in the morning and negative values in the afternoon that over the course of the diurnal cycle partially cancel the Twomey effect.*

TECHNICAL ISSUES:

Line 129: It seems that LWP refers to the sum of LWPc (cloud LWP) and RWP. If this is the case, please state it explicitly.

That interpretation is not correct. The LWP is defined on line 27 as the grid mean cloud liquid water path: LWP = $f_c$LWP$_c$. The old Line 129 (now 163) and the following equations explicitly reference the LWP$_c$. No changes are made to the manuscript as these definitions are already very clear.

Line 169: Should this refer to panel e instead of d? Please double-check.

Thanks for catching this typo We have changed d to e.

Since Section 2 is titled "Methodology," Subsection 2.4 should be moved to a new section for clarity and consistency in the manuscript structure.

Thanks for catching this. 2.4 and its subsections 2.4.1 – 2.4.3 have now been changed to section 3 with subsections 3.1. – 3.3. The conclusions is now section 4.

---

## Referee Report (RR1)

**Review of "The Diurnal Susceptibility of Subtropical Clouds to Aerosols" by Kurowski et al. (egusphere-2025-714)**

The authors addressed my previous questions and suggestions well. I only have a few minor suggestions, and support the manuscript's publication in Atmospheric Chemistry and Physics. I do not need to see the manuscript again.

Please not that line numbers refer to the tracked-changes version of the manuscript.

**Minor Comments**

Ll. 14 - 16: Based on II. 238 - 239, this statement should not refer to the adjustments (i.e., changes in LWP and fc, which are both negative in the afternoon) but the overall sensitivity, combining the Twomey effect and the aforementioned adjustments.

Ll. 245 – 246: Why is the Twomey effect constant? Typically, one assumes the Twomey effect to saturate for a higher cloud albedo, as expected for higher cloud droplet concentrations. What is the reason here?

**Technical Comments**

L. 35: "cloud optical thickness" to "\tau\_c"

L. 99: "cloud fraction" to "f\_c"

L. 140: "cloud droplet number concentrations" to "N\_c"

LL. 170 ff.: State the LWP, CRE, etc. using upright (non-italic) characters.

L. 244: Switch "N\_25" and "N\_50"

Fig. 1: "COT" to "\tau\_c", "Cloud fraction" to "f\_c", "CTH"

Fig. 3: Units in upright (non-italic) characters.

Figs. 2, 4, 5: Panel labels overlap with the ordinate's title.

---

## Author Response (AR2)

**RESPONSE TO THE EDITOR AND REVIEWERS**

Thank you to the Editor and the reviewers for their final comments. We have made minor changes to the manuscript to address the reviewer' comments, as explained below. The referee comments are in black, and our responses are in blue text.

The authors addressed my previous questions and suggestions well. I only have a few minor suggestions, and support the manuscript's publication in Atmospheric Chemistry and Physics. I do not need to see the manuscript again. Please not that line numbers refer to the tracked-changes version of the manuscript.

**Minor Comments**

Ll. 14 – 16: Based on II. 238 – 239, this statement should not refer to the adjustments (i.e., changes in LWP and fc, which are both negative in the afternoon) but the overall sensitivity, combining the Twomey effect and the aforementioned adjustments.

The text in I. 14-16 says: While the Twomey effect dominates the diurnal average albedo response, the diurnal variation in the competing cloud adjustments lead to a near-neutral net adjustment effect in the afternoon, highlighting the critical role of diurnally varying processes in aerosol-cloud interactions.

There must be some confusion here as we only say here that the net effect is near-neutral, not mentioning the sign of the remaining cloud adjustments.

Ll. 245 – 246: Why is the Twomey effect constant? Typically, one assumes the Twomey effect to saturate for a higher cloud albedo, as expected for higher cloud droplet concentrations. What is the reason here?

Good question. We now explain it in the text (l. 227):

The fact that the Twomey effect remains comparable for both aerosol regimes results from the fact that the strength of the effect is dominated during mid-day hours when all simulations have similar cloud albedos which are significantly smaller than unity (Fig. 2 c).

Technical Comments
L. 35: "cloud optical thickness" to "\tau\_c"

Changed.

L. 99: "cloud fraction" to "f\_c" Changed.

L. 140: "cloud droplet number concentrations" to "N\_c"

We think it is better to keep it as is in this particular place.

LL. 170 ff.: State the LWP, CRE, etc. using upright (non-italic) characters. Changed.

L. 244: Switch "N\_25" and "N\_50" Changed.

Fig. 1: "COT" to "\tau\_c", "Cloud fraction" to "f\_c", "CTH"

Changed. We left CTH as is as we are not sure what the suggestion was.

Fig. 3: Units in upright (non-italic) characters. Changed.

Figs. 2, 4, 5: Panel labels overlap with the ordinate's title Changed to fix the issue.